

# Spin and orbital spectroscopy in the absence of Coulomb blockade in lead telluride nanowire quantum dots

Maksim Gomanko[1], Eline J. de Jong[2], Yifan Jiang[1],
Sander G. Schellingerhout[2], Erik P. A. M. Bakkers[2] and Sergey M. Frolov[1*]

**1** Department of Physics and Astronomy, University of Pittsburgh, Pittsburgh, PA 15260, USA
**2** Eindhoven University of Technology, 5600 MB, Eindhoven, The Netherlands

⋆ frolovsm@pitt.edu

## Abstract

We investigate quantum dots in semiconductor PbTe nanowire devices. Due to the accessibility of ambipolar transport in PbTe, quantum dots can be occupied both with electrons and holes. Owing to a very large dielectric constant in PbTe of order 1000, we do not observe Coulomb blockade which typically obfuscates the orbital and spin spectra. We extract large and highly anisotropic effective Landé g-factors, in the range 20-44. The absence of Coulomb blockade allows direct readout, at zero source-drain bias, of spin-orbit hybridization energies of up to 600 $\mu$eV. These spin properties make PbTe nanowires, the recently synthesized members of group IV-VI materials family, attractive as a materials platform for quantum technology, such as spin and topological qubits.

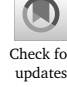

# 1 Introduction

Quantum computing relies on two-level systems (qubits) and state entanglement as leverage to attain computational power exceeding that of classical computers for certain classes of problems. Several qubit hardware realizations are being pursued at the moment, ranging from photons and trapped ions to nonlinear resonators, single charges and spins [1]. Single spins in low dimensional structures are natural candidates for qubits, and were one of the first platforms proposed [2]. Semiconductor quantum dots were used to perform single-shot spin state readout [3], coherent spin manipulation [4], and all-electrical spin control [5]. Spin qubits still have untapped potential due to their long spin coherence times, scalability and pathways for interfacing with conventional electronics via similar fabrication techniques, and relatively high operating temperatures [6].

Initial single spin experiments relied on heterostructures of group III-V semiconductors such as GaAs quantum wells [4], InAs and InSb nanowires [7–9]. More recently group IV semiconductors took center stage, in part due to reduced hyperfine interaction and longer potential coherence times [1]. Among realizations are Si/SiGe heterostructures [10], single atom donors in Si [11], Ge/Si core/shell nanowires [12], Ge quantum wells [13]. Another group IV family member, carbon, has been used in the nanotube form for valley-spin qubits [14].

PbTe and related materials were studied for applications such as infrared detectors and thermoelectrics [16] [17] [18]. They are characterized by the rocksalt crystal structure, low effective mass of $0.08m_e$, very large dielectric constant exceeding 1000 at low temperatures, small bandgap of 0.19 eV at 4.2K. Quantum wells of group IV-VI materials including PbTe exhibit quantum Hall effect and quantum point contact behavior [19–23]. Previous efforts to grow PbTe nanowires focused on chemical growth methods, such as chemical vapor deposition [16, 24] and hydrothermal synthesis [25, 26]. MBE growth has been reported [27] in Vapor-Liquid-Solid mode, though the nanowires were limited in length. A related material SnTe is a topological crystalline insulator [28], and has been explored in the nanowire form [29–32].

PbTe is composed of heavier elements, and is expected to have strong spin-orbit interaction [33], allowing electrical control of charge carrier spins, as well as large g-factors enabling low magnetic field operation of spin-orbit qubits [7]. The conduction band of PbTe is dominated by p-orbitals which has been associated with reduced hyperfine interaction [34]. In contrast with group III-V materials, where hyperfine coupling is the main decoherence mechanism for spin qubits, both Pb and Te have isotopes with zero nuclear spin, which can be leveraged through isotopic purification. Very large dielectric constants in PbTe suggest low charging energy of quantum dots, and screening of charge noise. Many of the these properties also make PbTe an interesting platform for the realization of Majorana zero modes, provided that superconducting proximity effect can be demonstrated [35].

We fabricate PbTe nanowire devices with electrical contacts in the 10kΩ range, as well as top and back gate electrodes. Quantum dots can be defined on both the electron and the hole side of the tunneling regime. Quantum dots exhibit no detectable charging energy. Coulomb blockade is absent due to a very large dielectric constant in PbTe. We extract g-factors as high as 44 with the anisotropy of order 2. We observe spin-orbit anticrossings between states of opposite spin belonging to different orbitals. The range of anticrossing values is from 10's to 100's of $\mu$eV indicative of sizeable spin-orbit coupling.

*Impact*. PbTe nanowires have recently been synthesized with low defect density and large aspect ratios, making them suitable for gate-defined quantum dot devices [36]. Absence of Coulomb blockade opens doors for the exploration of new regimes of spin manipulation and dynamics. This has enabled the investigation of spin, orbital and spin-orbit properties of electrons, as well as holes in this materials system. The IV-VI family features an impressive variety

of materials, from semiconductors and semimetals to topological crystalline insulators, and has significant potential for further exploration as a platform for quantum devices, such as spin and topological qubits.

## 2   Brief methods

PbTe nanowires are grown using molecular beam epitaxy (MBE) on GaAs[111] substrate in [100] axis orientation using Au catalyst [36]. Typical sizes are $1-1.5\,\mu$m in length and $70-125$ nm in width. We position the nanowires on doped silicon substrates with a 285 nm layer of thermal oxide. The substrate is used as a back gate. Metallic Ti/Au contacts (10/130 nm) are fabricated on top of the wire after argon sputter cleaning. Subsequently, the device is covered with a 10 nm $HfO_x$ layer by atomic layer deposition, and a top gate electrode (Ti/Au, 10/140 nm) is fabricated.

A standard low-frequency lock-in technique is used to acquire data. Measurements are performed in several dilution refrigerator setups, with base temperature of $50-100$ mK. Multiple devices were fabricated, varying in length and width (supplementary figure S1, panels (a), (c) and (d) have SEM pictures taken before covering the nanowire with a topgate). The major difference is that devices 5-8 did not had a topgate, only backgate separated by a wider dielectric layer. Device 1 is measured in a solenoid magnet with a fixed field orientation at 25 degrees to the nanowire axis. Device 2 is measured in a setup equipped with a 2D vector magnet with a 4T vector field that rotates in the substrate plane.

## 3   Figures

### 3.1   Figure 1: Device description

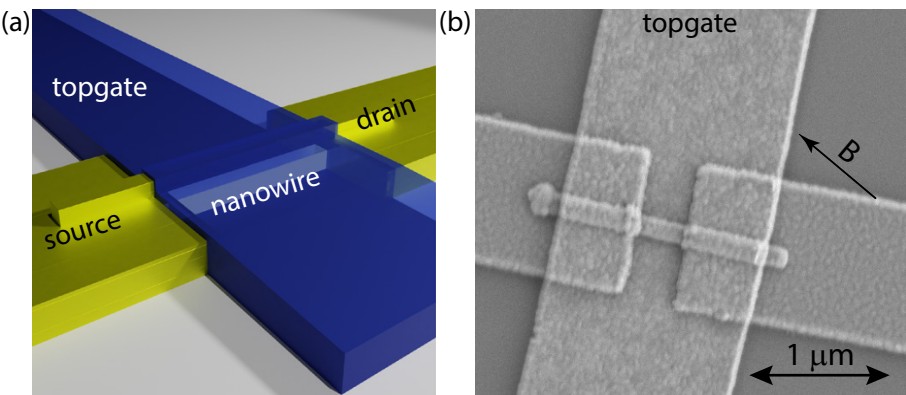

Figure 1: **a** Model of a topgate device. Contacts are colored with gold, top gate is represented by transparent blue. **b** SEM image of typical device (device 9). The orientation of magnetic field B corresponds to device 1 studied in Figs. 2 and 3.

Fig. 1 shows an example of a PbTe nanowire device. The spacing between the source and the drain is nominally 400 nm. PbTe nanowires have square cross-sections, in contrast with InAs and InSb nanowires which have hexagonal cross-sections. The nanowire width is in the range 70-125 nm. The top gate has a stronger effect due to it being closer to the nanowire and covering the nanowire on 3 sides. In fact the back gate alone is unable to pinch-off current in these devices (Supplementary figs. S10 and S10). Quantum dots are created between the

contacts, though their actual extent along the nanowire can be shorter.

## 3.2 Figure 2: Electron quantum dot

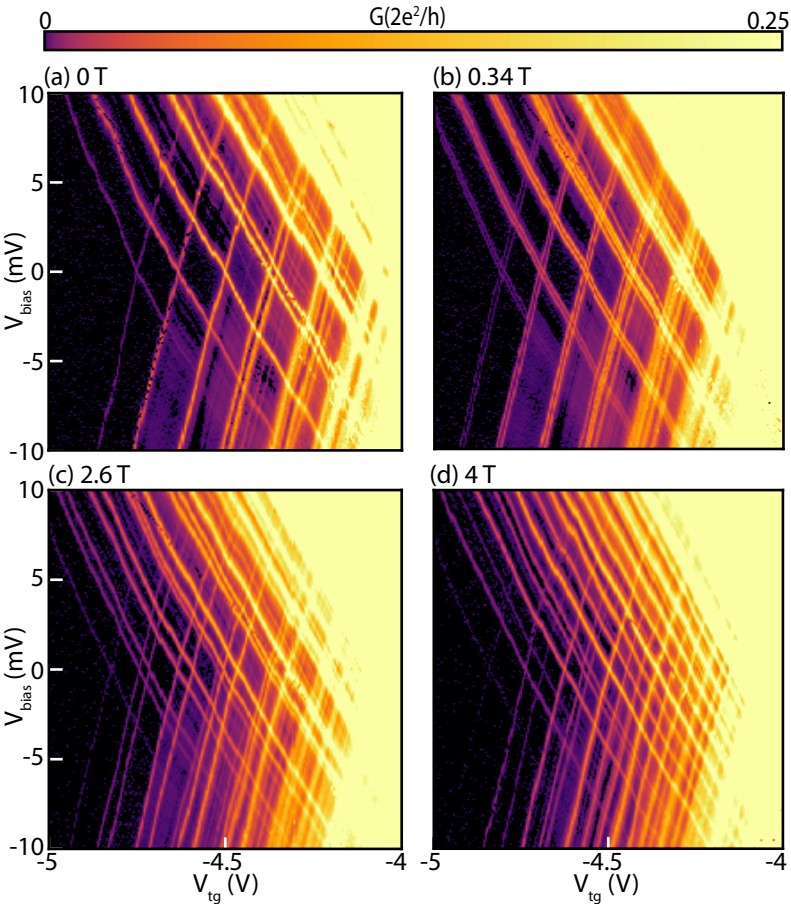

Figure 2: Differential conductance of device 1 as a function of $V_{bias}$ and $V_{tg}$ at different magnetic fields for $V_{bg}$ = -15V. **a** B = 0T; **b** B = 0.34T; **c** B = 2.6T; **d** B = 4.0T.

Figure 2 presents a study of an electron quantum dot. At zero magnetic field, we observe a grid of sharp intersecting conductance resonances (Fig. 2(a)). The pattern at first glance appears to be that of Coulomb blockade diamonds typical of quantum dots. However, upon closer inspection the data possess several unusual features.

First, there appear to be no excited states of the same total charge. These would be lines that run diagonally outside the central diamonds, and terminate at the edges of the diamonds. Coulomb blockade prevents the population of excited states of the same charge number - transport is blocked by an electron that is already present in the lowest energy state [37].

Second, at finite magnetic field each of the zero-field resonances splits (Figs. 2(b-d)). The splitting is apparent at B=0.34 T (panel b), while at a higher B=2.6 T we see a pattern of large/small diamonds around zero bias. At the highest field studied, B = 4 T, it becomes harder to track individual resonances but their density is doubled. The observation of resonance splitting in magnetic field is in contradiction with Coulomb blockade, which already lifts spin degeneracy at zero field.

Given the known large dielectric constant of PbTe we conclude that charging energy, and Coulomb blockade, are not observed in our devices. Related phenomena were reported in

LAO/STO quantum dots in the presence of superconductivity [38].

The apparent lack of resonances terminated at diamond edges becomes clear: in fact, all orbital and spin states are directly observed because they are not darkened by Coulomb blockade. They originate from consecutive orbital states at zero field, and spin-resolved states at finite field.

We note that at a given gate voltage, higher bias resonances do correspond to excited state transport when multiple electrons are allowed to travel through the dot simultaneously. This effect is the same with or without Coulomb blockade.

Since charging energy is negligible, from the height of central diamonds we can directly read off the orbital energies which are approximately 3 meV for the lowest few observable orbital states. While we cannot verify that the state at the most negative gate voltage corresponds to the first electron, we argue that the number of electrons in the quantum dot is a few, from the typical orbital energy and the gate voltage distance to the hole quantum dot on the valence band side (see Fig. 5).

At more positive gate voltages the resonances broaden gradually forming an open regime similar to Fabry-Perot and universal conductance fluctuation regimes (Figs. S10 and S10). The unusual aspect is that upon closing down the barriers no Coulomb blockade emerges, the resonances simply sharpen into grid of intersecting diagonals (Fig. 2).

### 3.3 Figure 3: Magnetic field spectroscopy

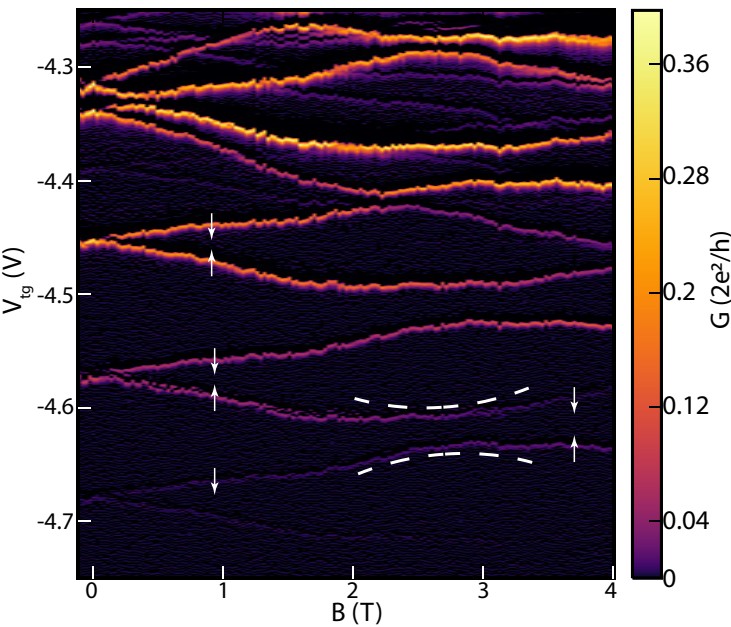

Figure 3: Differential conductance of device 1 as a function of topgate voltage and magnetic field. Arrows represent spin up and spin down states. Dashed curves illustrate an anticrossing of opposite-spin states. Magnetic field is applied at an angle of 25 degrees with respect to the nanowire axis, in the substrate plane.

Spin-resolved spectrum of the electron quantum dot in device 1 is presented in Fig. 3. The orbital states, which are visible as single resonances at zero magnetic field, undergo a splitting in finite magnetic field. Each orbital is doubly-degenerate due to spin, and no other degeneracies are detected. This is consistent with a lack of symmetry in the dot shape.

From the movement of the resonances in magnetic field, and the gate-bias lever arm extracted from data in Fig. 2, we can calculate the g-factors for three lowest visible orbitals to

be 28, 27, 29 (starting at the bottom).

Whenever resonances of opposite spin belonging to different orbitals come close, we observe level repulsion, which is the effect of spin-orbit coupling. The largest observed anticrossing is of the magnitude 575 $\mu$eV. Larger anticrossings tend to appear at higher magnetic fields which is consistent with spin-orbit induced hybridization getting enhanced by external field [39]. The ratios of spin-orbit to orbital energy vary between 0.28 and 0.10. This ratio approximately corresponds to the ratio between the quantum dot size and the spin-orbit length. Given the uncertainty in determining the quantum dot size, which can be a fraction of the contact spacing, we can only provide an upper bound estimate on the spin-orbit energy, though the effect is evident and strong.

We assume the effective mass 0.03$m_e$ at low temperatures [40]. We take the orbital energy $E_{orb} = 2.7 meV$ from Fig. 2. We estimate the size of the quantum dot from particle-in-a-box formula to be 80 nm, which is less than the contact spacing (350nm) and of the order of nanowire width (100nm). However, these numbers come with assumption of an isotropic box confinement, which is not the case for PbTe, and parabolic band structure, so a better theoretical evaluation is necessary. We obtain an estimate range for spin-orbit length of 300-900nm for different anticrossings.

While the conduction band of PbTe is known to have four-fold valley degeneracy [19], no degeneracies other than spin degeneracy manifest themselves in these quantum dots. Two possibilities exist. Either the valley degeneracy has been lifted, due to effects such as strain and strongly asymmetric quantum confinement. Alternatively, the valley degeneracy remains and is unperturbed by magnetic field, meaning that electrons in all valleys have exactly the same g-factors - and all observed resonances are in fact four-fold degenerate at finite field. The suppression of Coulomb blockade makes it difficult to distinguish between these possibilities.

Figure 3 also reveals faint resonances at more positive gate settings. Those resonances do exhibit Zeeman splitting, but they do not exhibit spin-orbit antricrossings with the bright resonances. We argue that those originate from another quantum dot given that their coupling to the leads is different and compounded by the fact that the dot is defined by a single wide gate - allowing for the possibility of other uncontrolled dots.

### 3.4 Figure 4: Effective Landé g-factor anisotropy

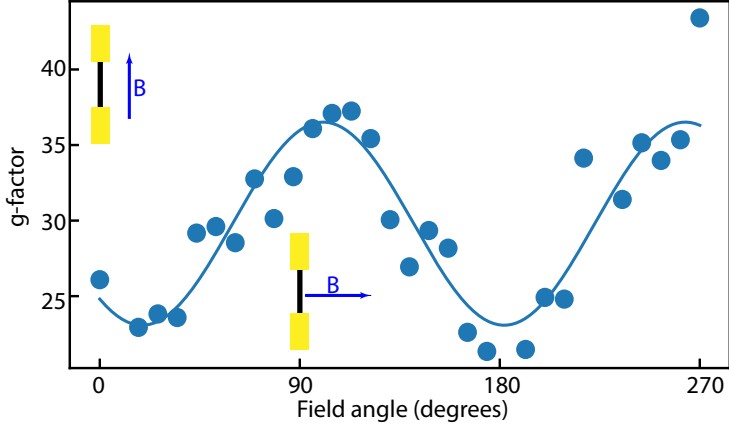

Figure 4: Effective Landé g-factor dependence on field angle for device 2. Graphics illustrate relative orientation of nanowire and magnetic field. Solid line is a sinusoidal fit with an offset of 29.8 and an amplitude of 6.7. Values extracted for state A in Fig.S6

.

We use device 2 to study the anisotropy of effective g-factors in a vector magnet setup. We fix magnetic field at the amplitude of 1T and rotate the field angle in the substrate plane. We extract g-factors from the height of the diamonds that emerge at finite field in between spin-up and spin-down states for each orbital, or from gate voltage shifts in peak positions at zero voltage bias. See supplementary information for details of data processing and for additional data.

In Fig. 4 we present the anisotropic g-factors for an orbital labeled A (Figs. S4-S6). The g-factor is lowest around zero angle, or along the nanowire, where it reaches the values of 22-24. The g-factor is largest when the field is perpendicular to the nanowire where it reaches 37.

From the g-factor anisotropy the shape of the quantum dot can be guessed. Enhanced g-factors originate from the coupling between spin and orbital states and are known to be reduced by quantum confinement that suppresses the orbital degrees of freedom [41]. The lower g-factors correspond to larger confinement in a direction perpendicular to the applied field. Based on these considerations, the dot is elongated along the nanowire which is in qualitative agreement with contact spacing being larger than the nanowire width.

The Fermi surface in PbTe is known to be anistoropic and this anisotropy can be contributing to the g-factor anisotropy as well as to other properties such as quantum dot level energies [42]. However, measurements done so far do not allow us to identify a contribution to the observed anisotropies or energies due to the crystal structure.

## 3.5   Figure 5: Hole quantum dot

In Fig. 5 that is also based on device 2 we present evidence of quantum states confined in the valence band. Hole quantum dots are a promising platform for quantum computing [13]. A handful of materials, such as carbon nanotubes and indium antimonide nanowires, make it possible to study hole and electron dots in the same sample, allowing to compare the orbital, spin and spin-orbit energies in different bands [14, 43].

In Fig. 5(a) we observe the transition from electron transport, across the bandgap, to hole transport which is characterized by significantly lower conductance. This is an interesting observation because the effective masses of electrons and holes in bulk PbTe are the same, thus similar mobility is expected. The discrepancy may come from differences in contact resistance due to different band character, or due to uncontrolled p-n junctions in the nanowire. Similar behavior has been observed in InSb nanowires [43]. Further investigation of this question will be required in locally gated devices and with the assistance of first principles theory.

Transport on the hole side is also dominated by resonances that criss-cross the gate-vs-bias diagram (Fig. 5(b)). Due to the lower currents the resonances are not seen in the same detail as on the electron side. However, they also do not exhibit Coulomb blockade. From the slopes and gate intervals separating the resonances, we estimate orbital energies in the range 2-4.5 meV. See supplementary information for additional valence band data on this and another device.

In a magnetic field we observe resonances splitting in a manner that is generally similar to how electron orbital states split. From various measurements we extract g-factors in the range 17-56 for apparent hole states. We do not observe spin-orbit level repulsion in these data but due to low currents we cannot conclude that spin-orbit effects are not present.

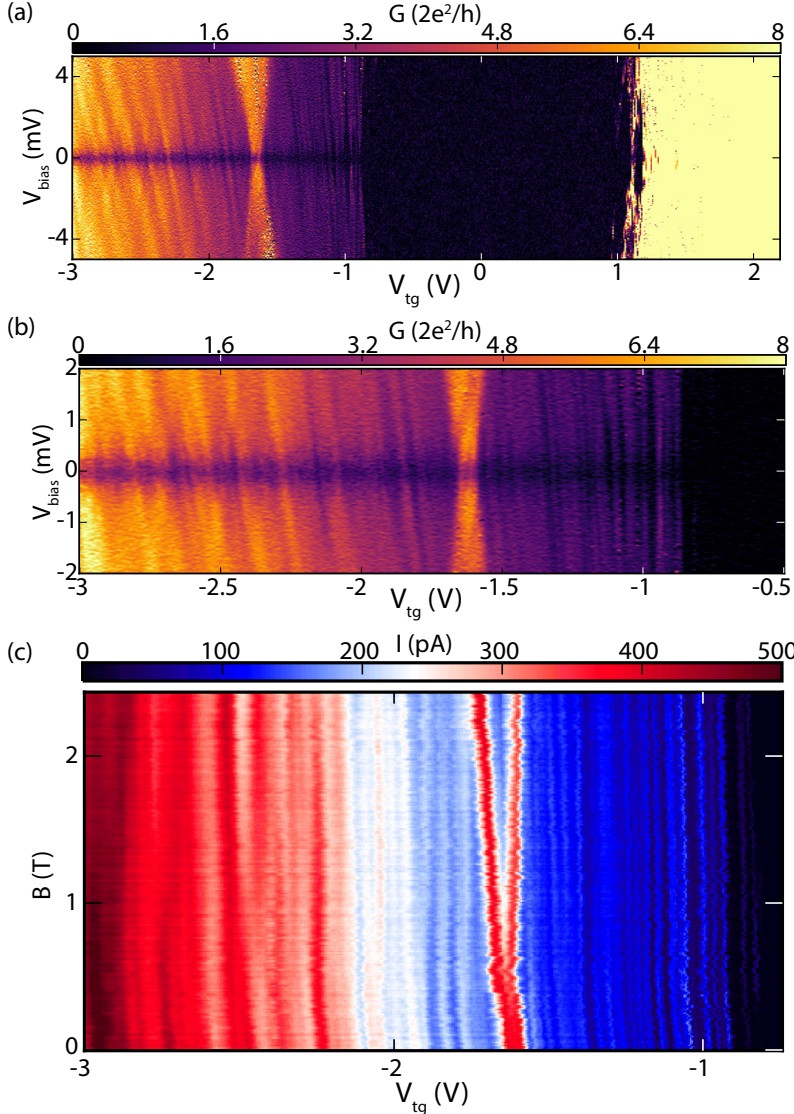

Figure 5: **a and b** Differential conductance for device 2. **c** Magnetic field vs topgate on the hole side. Magnetic field is at 45 degrees with respect to the nanowire.

## 4   Future work

In future experiments, multiple local gates can be used to define quantum dots of fixed length and provide an accurate estimate of spin-orbit interaction strength, to establish the double quantum dots and the few-electron/hole regime. This will also provide better control of dot barriers and thus higher resolution spectroscopy on the valence band side.

PbTe quantum dots free of charging energy have potential to become a promising platform for spin qubits. It would be interesting to explore whether large dielectric translates into longer spin coherence times. Another future question is whether Pauli spin blockade, an established spin qubit initialization and readout mechanism, can be observed in the absence of Coulomb blockade. Large anisotropic g-factors and strong spin-orbit interaction are promising for electric field control of spin states and for the operation of spin-orbit qubits.

Integration with superconductors, such as native and lattice-matched Pb, can make possible the integration of PbTe wires into superconducting qubits [44], as well as their exploration as a platform for Majorana zero modes [35].

## 5 Duration and Volume of Study

Experiments in this paper were done over a 1.5 year timeframe, including a lab shutdown due to the pandemic. A total of 20 chips with an average of 7 devices per chip were fabricated, with 50% not resulting in any data due to fabrication issues, contact barriers, gate leaks or inability to pinch them off. In this paper we use 900 datasets from 4 different chips, one of them with devices lacking a topgate. Quantum dot features were observed in 5 devices, clearest 3 of which were studied thoroughly and presented in this paper. The first chip contains devices 1 (Figs. 2,3), 3 (Fig. 5) and 4. The second chip contains device 2 (Figs. 4, 5). The third chip, without a topgate, has 4 devices (5,6,7,8) used in Figs. S9 and S10. Finally, the fourth chip was used for the SEM picture in Fig. 1. The numbering of chips is not chronological.

## 6 Data and Code availability

All of the data from devices used in this paper along with experimental logs, relevant notebooks and overview summaries is available on Zenodo (DOI 10.5281/zenodo.5531917). The repository also includes Python notebooks used for data analysis, e.g. to extract the g-factors from the data.

## A Supplementary Information

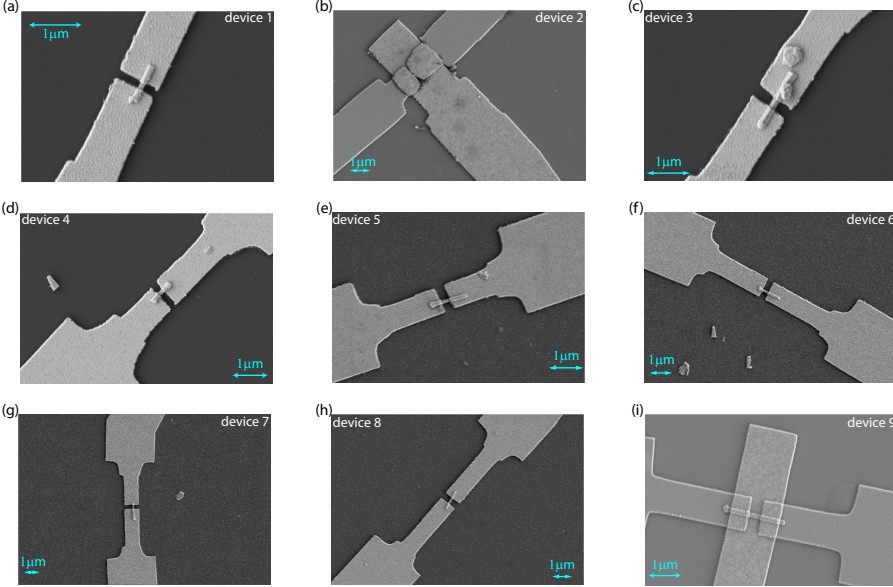

Figure S1: Scanning electron microscope images of all devices used in this paper. (a) device 1, used in Figs. 2 and 3 before fabrication of the topgate, (b) device 2, liftoff issues with topgate made nanowire hard to see, (c) device 3 before fabrication of the topgate, (d) device 4, (e)-(h) backgate chip, devices 5-8; (i) device 9, used for Fig. 1

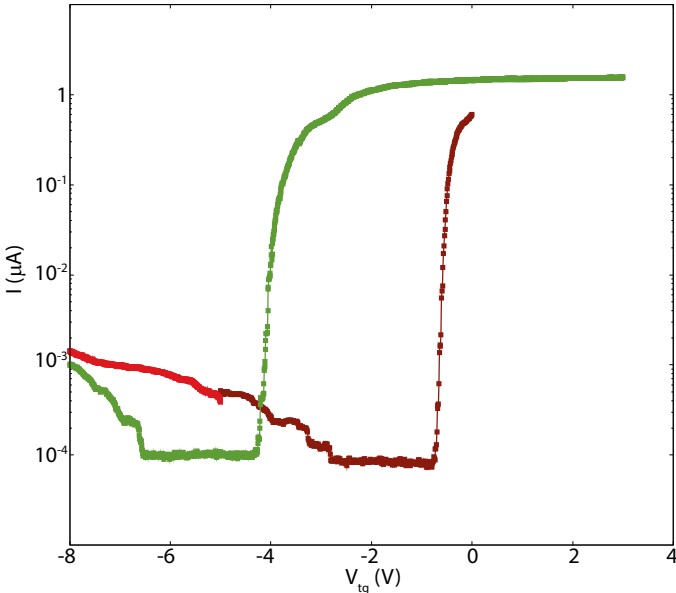

Figure S2: Bipolar transport in device 3. Global backgate voltage = -30V, B = 0T, $V_{bias}$ = 10mV. An offset of 0.2 nA was added for log-scale plotting. Three different colors represent 3 different datasets, taken one after another. Red colors were taken with increase in topgate voltage, while green color line represents decreasing voltage. Gate hysteresis is typical for large-range gate sweeps.

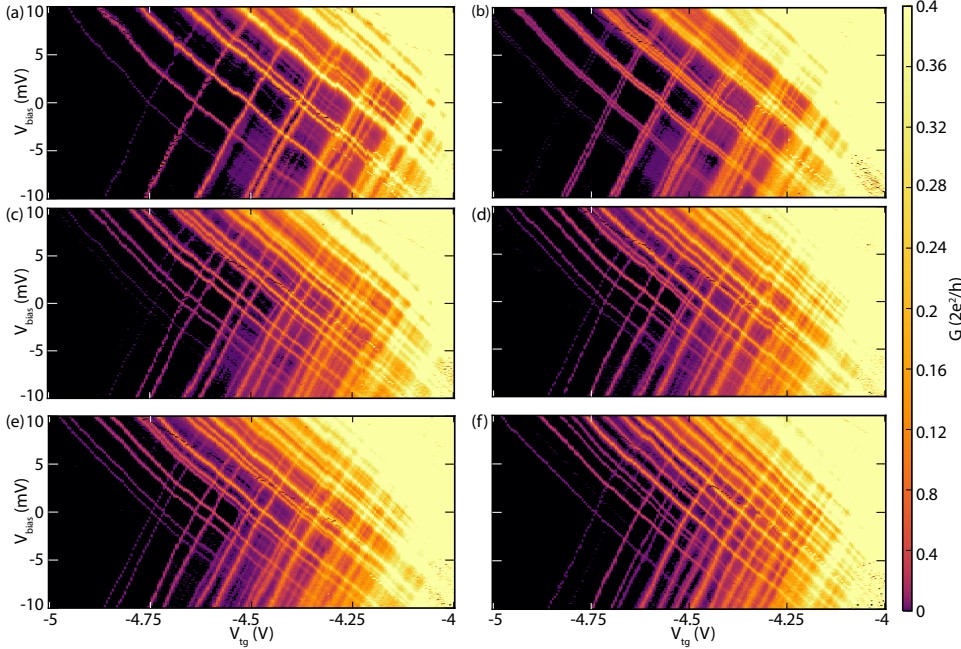

Figure S3: Additional data for Figure 2 from device 1. Global backgate voltage = -15V. (a) 0T, (b) 0.34T (c) 1T (d) 2.2T (e)2.6T (f)4.0T. Magnetic field is pointing at a 25 degree angle relative to the nanowire axis.

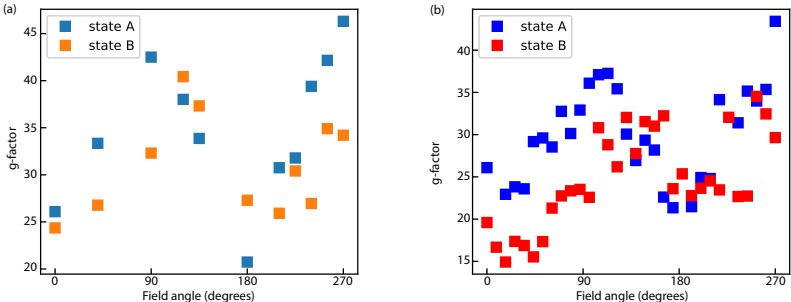

Figure S4: Two methods for calculating effective g-factors: **a** directly extracting Zeeman energy at B=1T the height of diamonds A and B in Fig. S5; **b** From data in Fig. S5 using the lever arm between gate voltage and bias voltage from Fig. S5. Data for state A are used in Fig. 4. States A and B are labeled in Figs. S4 and S5 and correspond to adjacent orbitals of the same quantum dot. The two methods yield qualitatively the same results but disagree quantitatively largely due to charge instabilities seen in Fig. S5.

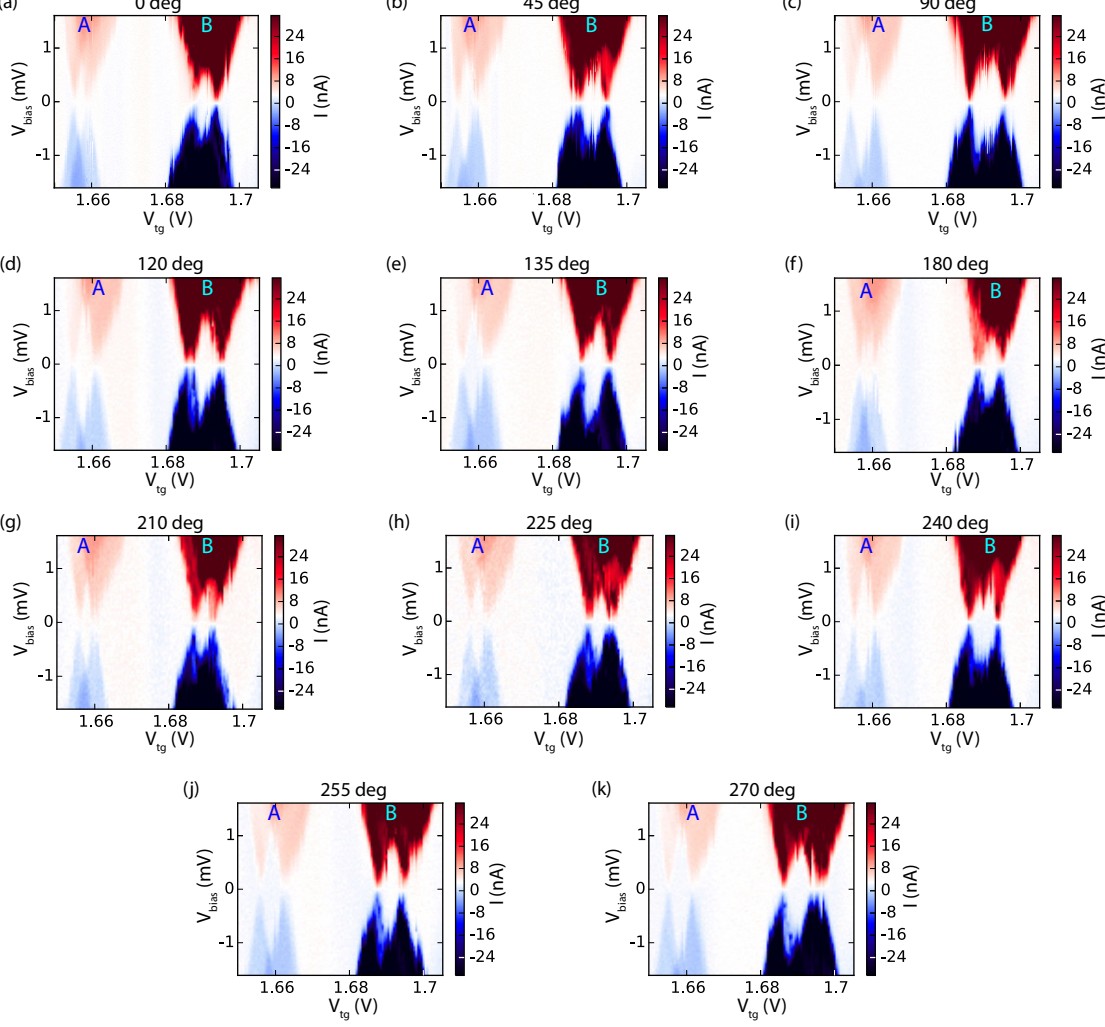

Figure S5: Measurements of current as function of $V_{bias}$ and $V_{tg}$ for device 2. Different panels correspond to different magnetic field orientations indicated above each panel with respect to the nanowire axis. The magnetic field amplitude is 1 Tesla. States A and B are labeled. Global backgate voltage is zero.

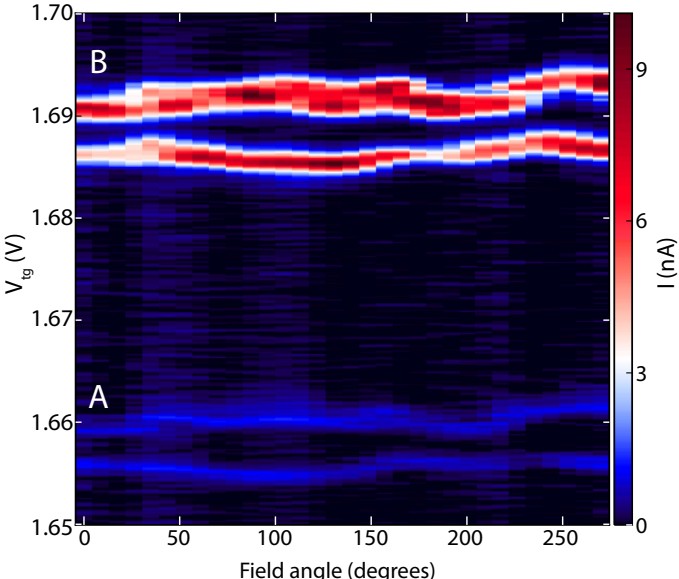

Figure S6: Device 2. Two lowest resolved electron QD states A and B as a function of topgate and magnetic field angle. State A is on the bottom, state B is on top of the figure. B=1T, $V_{bias} = 0.1$mV, global backgate voltage is zero.

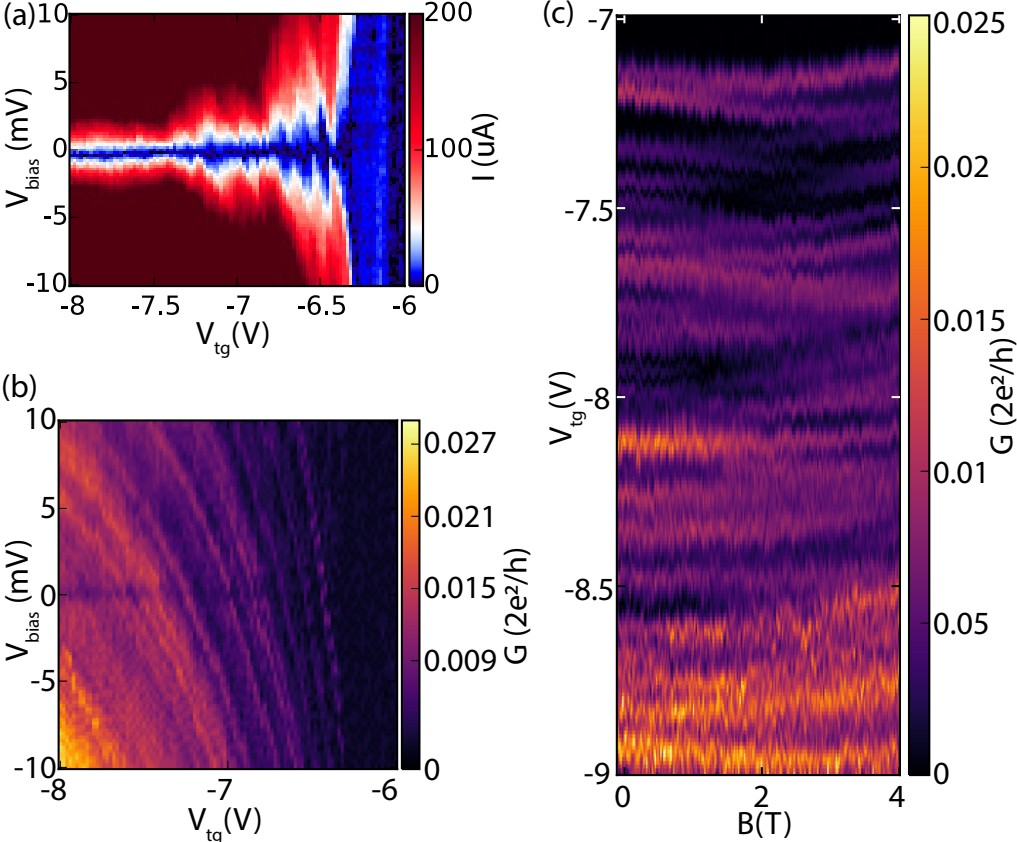

Figure S7: Hole states in device 3. (a) and (b) absolute value of current and differential conductance of the hole quantum dot, both panels are derived from the same dataset. B = 0. (c) Magnetic field evolution at $V_{bias}$=0.1mV Global backgate voltage is zero.

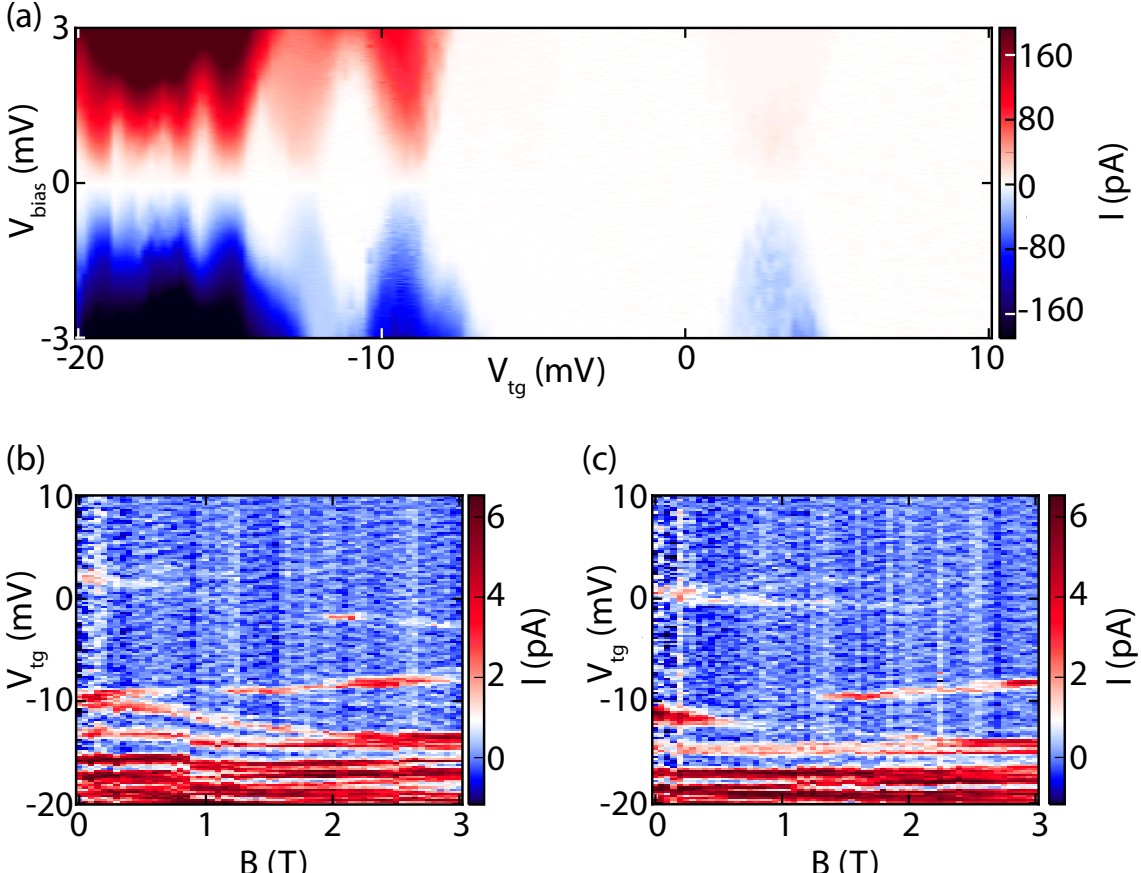

Figure S8: Hole quantum dot in device 2. Different state of the device can be explained by charge jumps occuring after Fig.5 was measured. (a) Current through the nanowire on the valence band side. It is possible that states at $V_{tg}$=-12 mV and 3 mV originate from another quantum dot that has smaller coupling to contact leads. (b) Magnetic field sweep at $V_{bias}$=0.1mV with field parallel to nanowire axis and (c) with field perpendicular to the nanowire axis. The two states mentioned above show stronger Zeeman splitting than other states in the picture that is comparable to the Zeeman splitting for the states on the conduction band side. We therefore hypothesize that these two states are from a different quantum dot, which may contain electrons rather than holes.

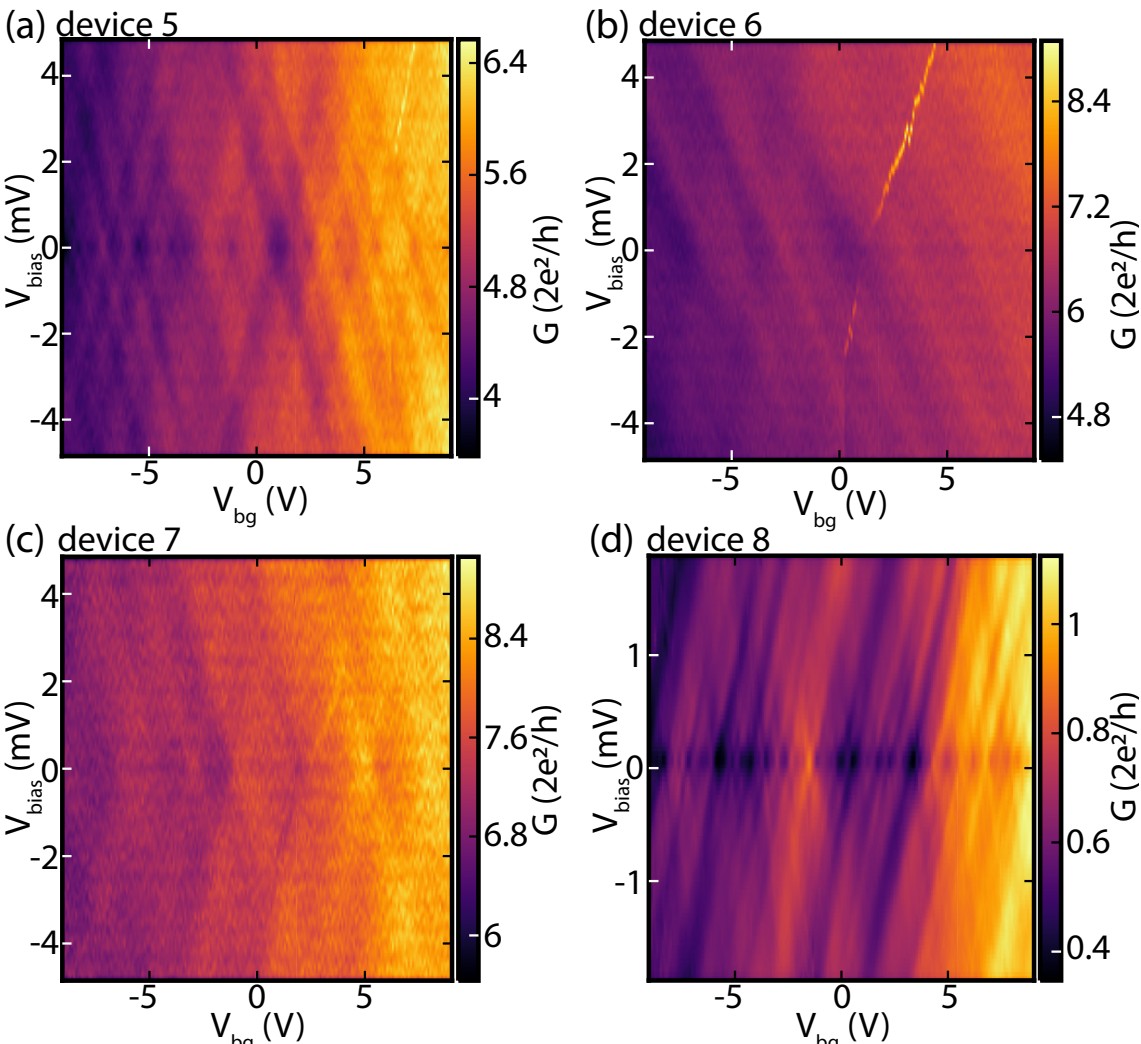

Figure S9: Data from four devices fabricated without a top gate, and only tunable by a back gate. In these devices it was not possible to reach the pinched-off regime with the chemical potential in the bandgap, all data are on the electron side. **a** is from device 5, **b** is from device 6, **c** is from device 7, **d** is from device 8. In the open regime we see a picture similar to quantum dot behavior in topgate devices, but with broadened resonances. They can be interpreted as Fabry-Perot modes in the nanowire, but evolve continuously into single orbital quantum levels such as in Fig. 2 when the gate potentials are decreased, and the resonances sharpen. B = 0T. Conductance is calculated by numerically diffentiating DC current evolution in bias voltage.

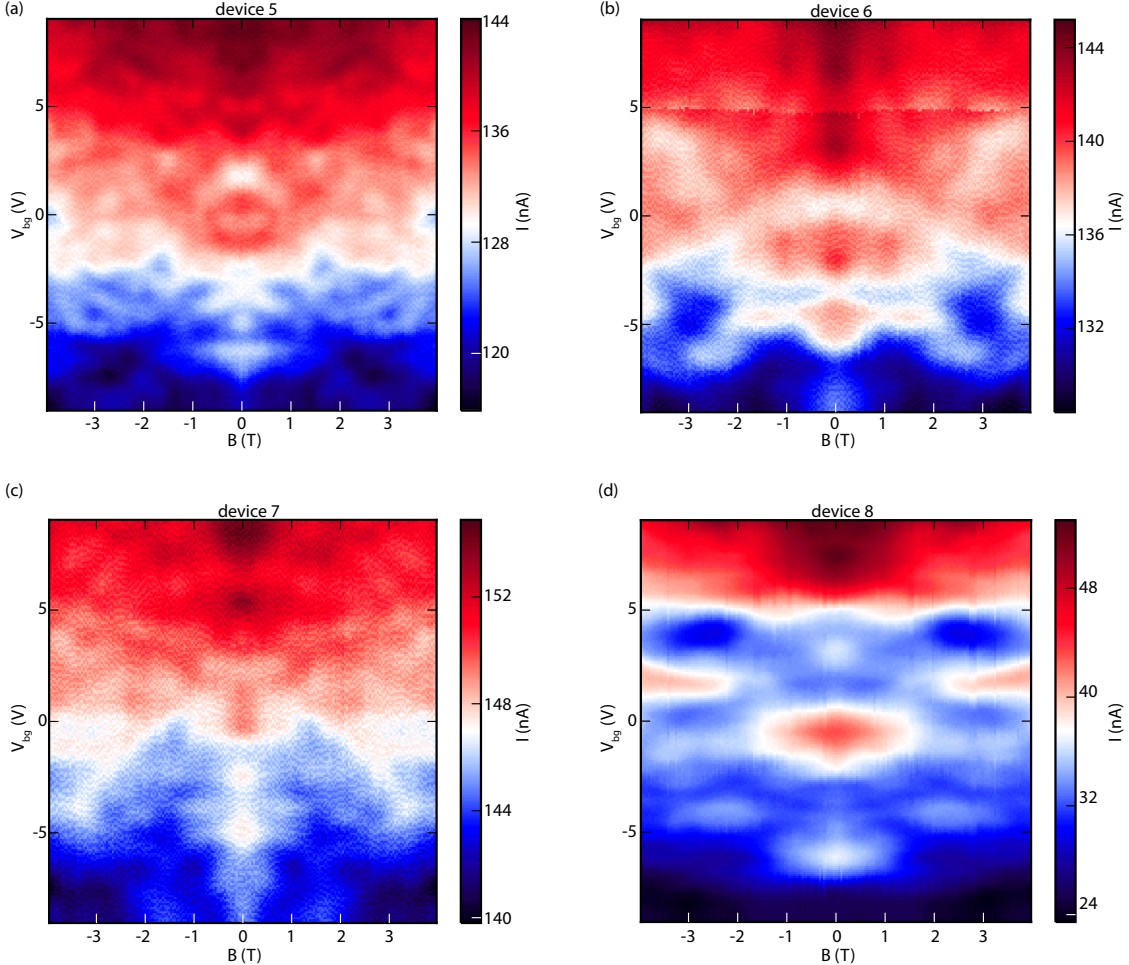

Figure S10: Magnetic field evolution of conductance in devices without a top gate shown in Fig. S9. (**a** is from device 5, **b** is from device 6, **c** is from device 7, **d** is from device 8. $V_{bias}$=1mV, magnetic field direction varies for every device. It is applied horizontally with respect to images in Fig. S1, resulting in (a) 14, (b) 63, (c) 86, (d) 58 degree angle. G-factor in **d** (device8) for state at $V_{bg}$=-5V is estimated to be 11 after converting backgate voltage to bias using lever arm from data in Fig. S10 **d**.

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
