# Peer review of "Spin and Orbital Spectroscopy in the Absence of Coulomb Blockade in Lead Telluride Nanowire Quantum Dots"

_SciPost Physics, doi:SciPost Phys. 13, 089 (2022)_

## Round 1 · Referee Report · Anonymous (Referee 1) · 2022-2-3

Strengths

  • The authors carry out mesoscopic measurements on PbTe nanowire quantum dots and discuss their potential for quantum devices. PbTe can have strong spin-orbit splitting and has a large g-factor, making it indeed attractive for the application that the authors consider. It is actually surprising that this material has not yet been considered for this research direction.
  • The strength and anisotropy of the g-factor are measured for the first time in such structures and argue that PbTe can compete with III-Vs.
  • The type of measurements combined with the analysis that is carried out are novel and can lead to progress and more questions in an important research area. This is especially true given the fact that the dielectric constant of PbTe is high, which has both advantages and disadvantages when it comes to quantum device performance as pointed out by the authors.
  • A sufficient number of devices have been studied and are reported on.

Weaknesses

  • Some of the analysis lacks depth. The anisotropy of the g-factor is not discussed in the context of the band structure of PbTe and the anisotropy of its Fermi surface.

  • The valley degeneracy of PbTe is ignored but the argument behind why that is done is not sufficient.

Report

The authors report mesoscopic transport measurements done on PbTe nanowire quantum dots. The measurements allow the extraction of the spin-orbit coupling strength and the g-factor of PbTe. The measurements are well presented and the results are important as they demonstrate the potential of PbTe compared to III-V materials for semiconductor-superconductor hybrids. This is especially important for quantum computing devices and the search for Majorana modes. I think the manuscript is suitable for this journal and should be considered for publication based on criterion 3 for acceptance: Open a new pathway in an existing or a new research direction, with clear potential for multipronged follow-up work.

But before it can be accepted, the authors should address the issues listed below, in the requested changes.

Requested changes

  1. The Fermi surface of PbTe is inherently anisotropic (non spherical) and valley degenerate. This fact is discussed in several reports on single crystals and films (Melngailis et al. PRB 3 370 1971, Bauer Narrow Gap Semiconductors Physics and Applications: Proceeding of the International Summer School 133 427 (1980) , Hayasaka J. Phys.: Condens. Matter 28 (2016) 31LT01 ). Does that have any impact on the energy levels of the quantum dot and on the g-factor?

  2. Related to that, what is the crystallographic direction of the long axis of the nanowire? This is important given the highly anisotropic character of the electronic structure of PbTe.

  3. In Fig. 3, could the faint resonances observed at high Vtg be due to lifting of valley degeneracy instead of a separate quantum dot?

  4. Can the authors at least mention the limits of an isotropic particle-in-a-box model for PbTe? I am pointing this out both because the Fermi surface of PbTe is anisotropic and because its band structure is highly non-parabolic and approaches a Dirac-like structure at low energies. (see for example Phys. Rev. B 98, 115144 (2018) or NPJ Quantum Material 2 26 (2017)). While I can accept that they limit the discussion in this simple case to a basic model to estimate the size of the quantum dot, a mention of this limitation can at least encourage more elaborate theoretical efforts to compute the electronic structure of these dots.

  5. Can the authors mention the temperature at which each measurement is carried out? How robust is their signal to temperatures approaching 4.2K? Grabecki et al report conductance quantization even at 1.8K ref[20].

  6. Some minor things:

  7. For the dielectric constant of PbTe the authors can reference Preier Applied Physics 20 189 (1980).

  8. A reference is needed when the authors mention the valley degeneracy of PbTe. The current version has [?] as the reference.

  9. When mentioning the topological phase in IV-VI materials is mentioned, the authors should give credit to the following experimental work: on Dziawa et al. Nature Materials 11, 1023 (2012), and Tanaka et al Nature Physics 8 800 (2012).

  • validity: high
  • significance: top
  • originality: top
  • clarity: high
  • formatting: excellent
  • grammar: perfect

Author:  Sergey Frolov  on 2022-05-24  [id 2525]

(in reply to Report 1 on 2022-02-03)

Referee 1:

Weaknesses - Some of the analysis lacks depth. The anisotropy of the g-factor is not discussed in the context of the band structure of PbTe and the anisotropy of its Fermi surface.

Response: We do not have evidence to claim that those anisotropies play a role. The observed g-factor anisotropy is consistent with confinement anisotropy (dots elongated along the nanowire, which is reasonable for dots defined between contacts). We added a mention of the other factors to the text with a statement that while they may play a role, we do not have data to separate them out.

  • The valley degeneracy of PbTe is ignored but the argument behind why that is done is not sufficient.

Response: So far, we do not see evidence of valley degeneracy in these measurements. In a magnetic field, resonances split in two consistent with Zeeman splitting. It could be that each of those resonances actually corresponds to multiple electrons from different valleys. But then g-factors would be identical in all of the valleys, so that valley degeneracy is maintained at high field.

In quantum dots with Coulomb energy, valley degeneracy is relatively easy to establish, for instance in carbon nanotube quantum dots three small diamonds followed by a large diamond signify two-fold valley degeneracy. In quantum dots with quenched Coulomb interaction, we cannot see this. But we may be able to identify other means of understanding what happens to the valley degree of freedom in these devices in subsequent measurements.

We added this discussion to the manuscript.

Report

The authors report mesoscopic transport measurements done on PbTe nanowire quantum dots. The measurements allow the extraction of the spin-orbit coupling strength and the g-factor of PbTe. The measurements are well presented and the results are important as they demonstrate the potential of PbTe compared to III-V materials for semiconductor-superconductor hybrids. This is especially important for quantum computing devices and the search for Majorana modes. I think the manuscript is suitable for this journal and should be considered for publication based on criterion 3 for acceptance: Open a new pathway in an existing or a new research direction, with clear potential for multipronged follow-up work. But before it can be accepted, the authors should address the issues listed below, in the requested changes. Requested changes 1. The Fermi surface of PbTe is inherently anisotropic (non spherical) and valley degenerate. This fact is discussed in several reports on single crystals and films (Melngailis et al. PRB 3 370 1971, Bauer Narrow Gap Semiconductors Physics and Applications: Proceeding of the International Summer School 133 427 (1980) , Hayasaka J. Phys.: Condens. Matter 28 (2016) 31LT01 ). Does that have any impact on the energy levels of the quantum dot and on the g-factor?

Response: quantum dots are 0-dimensional, so there is no defined k. On top of that, confinement is typically anisotropic, and we believe it is here, based on g-factor anisotropy. Confinement anisotropy determines the anisostropy of various parameters such as the g-tensor. It also leads to band and valley mixing, and orbital mixing (manifesting in spin-orbit anicrossings). On top of these effects, it is challenging to impossible to resolve the features due to the anisotropy of the underlying Fermi surface. It may be possible in a large study where confinement is controlled, nanowire growth direction is varied etc.

We added a discussion of this point and the references the referee pointed to.

  1. Related to that, what is the crystallographic direction of the long axis of the nanowire? This is important given the highly anisotropic character of the electronic structure of PbTe.

Response: [100], Added this information to the “brief methods” section.

  1. In Fig. 3, could the faint resonances observed at high Vtg be due to lifting of valley degeneracy instead of a separate quantum dot?

Response: It is possible that different valleys are shifted by an energy greater than several orbital energies. Though in that case, we expect simply filling the same dot with electrons belonging to different valleys, just like electrons of a different spin. But with an offset, so that the first few electrons all belong to one valley. Here we observe fainter resonances that do not interact with the more visible set at more negative Tg.

It is more likely that several quantum dots are created in the nanowire. In future measurements, we shall use local gates that would allow us to define quantum dots, including multiple dots, with greater control and distinguish between these possibilities.

  1. Can the authors at least mention the limits of an isotropic particle-in-a-box model for PbTe? I am pointing this out both because the Fermi surface of PbTe is anisotropic and because its band structure is highly non-parabolic and approaches a Dirac-like structure at low energies. (see for example Phys. Rev. B 98, 115144 (2018) or NPJ Quantum Material 2 26 (2017)). While I can accept that they limit the discussion in this simple case to a basic model to estimate the size of the quantum dot, a mention of this limitation can at least encourage more elaborate theoretical efforts to compute the electronic structure of these dots.

Response: We added a mention of the limitations of this approach to the paper. “However, these numbers come with an assumption of isotropic box confinement, which is not the case for PbTe, and parabolic band structure, so a better theoretical evaluation is necessary.”

  1. Can the authors mention the temperature at which each measurement is carried out? How robust is their signal to temperatures approaching 4.2K? Grabecki et al report conductance quantization even at 1.8K ref[20].

Response: We added these details to the paper: : A standard low-frequency lock-in technique is used to acquire data. Measurements are performed in several dilution refrigerator setups, with a base temperature of 50-100 mK.

We did not perform any measurements at higher temperatures, except for quick 1D bias and gate sweeps for backgate-only devices (5-8) to check IV curves and backgate voltage dependence. There was no study of these nanowires at higher temperatures except for wire resistance at room temperature to choose devices for cooldown.

It is not surprising to see conductance quantization at higher temperatures. In fact it should get more clear once coherence length is suppressed by temperature and conductance fluctuations due to quantum interference are reduced in amplitude. The energy scale of quantized subbands is ~1meV which is 10K.

  1. Some minor things:
  2. For the dielectric constant of PbTe the authors can reference Preier Applied Physics 20 189 (1980).

We feel that the large dielectric constant of PbTe is widely known and does not require citation. That paper itself references other papers (Dalven, infrared physics, 1969), with the low-frequency dielectric constant being 428 calculated from high-frequency one and optical measurements of 200-400.

  • A reference is needed when the authors mention the valley degeneracy of PbTe. The current version has [?] as the reference.

Response: Fixed now. Thanks!

  • When mentioning the topological phase in IV-VI materials is mentioned, the authors should give credit to the following experimental work: on Dziawa et al. Nature Materials 11, 1023 (2012), and Tanaka et al Nature Physics 8 800 (2012).

Response: We reference these papers in the revised manuscript.

---

## Round 1 · Referee Report · Anonymous (Referee 2) · 2022-2-9

Strengths

  1. Interesting material (PbTe narrow bandgap semiconductor).
  2. MBE-grown nanowires as the objects of magnetotransport measurements.
  3. Back-gate voltage dependent transport measurements, quite unique for PbTe.
  4. Notion of the absence of Coulomb blockade effects enabling extraction of the spin-orbit hybridization energies.

Weaknesses

  1. Introductory part does not refer comprehensively to the existing literature concerning IV-VI narrow bandgap semiconductor nanowires.
  2. The basic parameters of PbTe NWs used for the measurements - axes orientation, lengths diameters are not specified.
  3. The differences between individual devices characterized by magnetotransport measurements are not specified.

Report

The authors report on back-gate voltage dependent magnetotransport of single PbTe nanowires.
They notice the absence of Coulomb blockade effects, due to the high static dielectric constant of PbTe, which allows extracting the spin-orbit hybridization energies.
The paper is scientifically sound and can be published after minor revision; only some doubts concerning the introductory part and some minor details in the main text arise, as specified below.

Requested changes

  1. In the introductory part, the authors write; Previous efforts to grow PbTe nanowires focused on chemical vapor deposition [21, 22]. A related material SnTe is expected to be a topological crystalline insulator, and has been explored in the nanowire form [23, 24].

1a. The authors ignore quite numerous literature reports concerning PbTe nanowires fabricated by chemical methods such as hydrothermal synthesis: Guoan Tai, Wanlin Guo, and Zhuhua Zhang, Cryst. Growth & Design 8, 2906 (2008);
Qingyu Yan, et al., Chem. Mater. 20, 6298, (2008);

and by MBE: P. Dziawa, et. al., Cryst. Growth and Design, 10, 109 (2010)

The papers reporting on transport and thermoelectric properties of PbTe NWs also deserve citing: e.g.: So Young Jang, et. al., Transport properties of single-crystalline n-type semiconducting PbTe nanowires. Nanotechnology 20, 415204, (2009) Jong Wook Roh, et. al., Size-dependent thermal conductivity of individual single-crystalline PbTe nanowires. Appl. Phys. Lett. 96, 103101 (2010).

1b. Further in the introductory part the authors state that: A related material SnTe is expected to be a topological crystalline insulator, and has been explored in the nanowire form [23, 24].

SnTe is not an “expected” but proved crystalline topological insulator, which is evidenced e.g., here: Y. Tanaka, et al., Experimental realization of a topological crystalline insulator in SnTe. Nat. Phys. 8, 800 (2012). Also references 23, and 24 do not grasp the already published reports on SnTe NWs, especially in the context of SnTe NWs grown by MBE: J. Sadowski, et al., Defect-free SnTe topological crystalline insulator nanowires grown by molecular beam epitaxy on graphene. Nanoscale, 10, 20772, (2018).

  1. The first phrase of the section “Brief methods” reads: PbTe nanowires are grown using molecular beam epitaxy (MBE) [28]

Further in the text there is no information neither on the geometrical parameters (lengths diameters) nor on crystalline orientation of NWs, what is the NW axis direction? It can only be deduced that it is [100] since in one place further in the text the authors mention that the NWs have square cross-sections. Sending the reader to the paper published already - Ref. [28] is not fair; the reader cannot be forced to dig into the previous papers published by the authors, to understand the current one.

  1. On the top of the page 2 the authors introduce Device 1 and 2 without specifying the difference between them.

  2. Further on the authors refer to Figs. S9 and S10, without specifying that these figures are placed in the supplementary material.

  3. On page 2, right column lines 7-8 for the bottom: While the conduction band of PbTe is known to have four-fold valley degeneracy [? ] Do the authors have doubts about the validity of this phrase, or they have forgotten to specify the appropriate reference?

  • validity: high
  • significance: high
  • originality: top
  • clarity: good
  • formatting: good
  • grammar: excellent

Author:  Sergey Frolov  on 2022-05-24  [id 2526]

(in reply to Report 2 on 2022-02-09)

Weaknesses 1. Introductory part does not refer comprehensively to the existing literature concerning IV-VI narrow bandgap semiconductor nanowires.

Response: we expanded our literature overview.

  1. The basic parameters of PbTe NWs used for the measurements - axes orientation, lengths diameters are not specified.

Response: We have now specified these parameters.

  1. The differences between individual devices characterized by magnetotransport measurements are not specified.

Response: We explained this now. There are two types of devices: with a top gate and without, and most measurements are on top-gate devices. They are nominally the same, though the nanowires are different, including their diameters, as well as contact spacings.

Requested changes 1. In the introductory part, the authors write;

Previous efforts to grow PbTe nanowires focused on chemical vapor deposition [21, 22]. A related material SnTe is expected to be a topological crystalline insulator, and has been explored in the nanowire form [23, 24].

1a. The authors ignore quite numerous literature reports concerning PbTe nanowires fabricated by chemical methods such as hydrothermal synthesis: Guoan Tai, Wanlin Guo, and Zhuhua Zhang, Cryst. Growth & Design 8, 2906 (2008); Qingyu Yan, et al., Chem. Mater. 20, 6298, (2008); and by MBE: P. Dziawa, et. al., Cryst. Growth and Design, 10, 109 (2010) The papers reporting on transport and thermoelectric properties of PbTe NWs also deserve citing: e.g.: So Young Jang, et. al., Transport properties of single-crystalline n-type semiconducting PbTe nanowires. Nanotechnology 20, 415204, (2009) Jong Wook Roh, et. al., Size-dependent thermal conductivity of individual single-crystalline PbTe nanowires. Appl. Phys. Lett. 96, 103101 (2010). 1b. Further in the introductory part the authors state that: A related material SnTe is expected to be a topological crystalline insulator, and has been explored in the nanowire form [23, 24]. SnTe is not an “expected” but proved crystalline topological insulator, which is evidenced e.g., here: Y. Tanaka, et al., Experimental realization of a topological crystalline insulator in SnTe. Nat. Phys. 8, 800 (2012). Also references 23, and 24 do not grasp the already published reports on SnTe NWs, especially in the context of SnTe NWs grown by MBE: J. Sadowski, et al., Defect-free SnTe topological crystalline insulator nanowires grown by molecular beam epitaxy on graphene. Nanoscale, 10, 20772, (2018).

Response: Thanks for pointing it out, we reference these papers in the revised manuscript.

  1. The first phrase of the section “Brief methods” reads: PbTe nanowires are grown using molecular beam epitaxy (MBE) [28] Further in the text there is no information neither on the geometrical parameters (lengths diameters) nor on crystalline orientation of NWs, what is the NW axis direction? It can only be deduced that it is [100] since in one place further in the text the authors mention that the NWs have square cross-sections. Sending the reader to the paper published already - Ref. [28] is not fair; the reader cannot be forced to dig into the previous papers published by the authors, to understand the current one.

Response: We edited the “brief methods” section so there will be more details about our nanowires growth ([100] axis on GaAs[111] substrates and their dimensions.

  1. On the top of the page 2 the authors introduce Device 1 and 2 without specifying the difference between them.

We also added a comment on differences between our devices and reference supplementary figure S1 that illustrates them.

  1. Further on the authors refer to Figs. S9 and S10, without specifying that these figures are placed in the supplementary material.

Fixed, now it specifies that these figures are in the supplementary section.

  1. On page 2, right column lines 7-8 for the bottom: While the conduction band of PbTe is known to have four-fold valley degeneracy [? ] Do the authors have doubts about the validity of this phrase, or they have forgotten to specify the appropriate reference?

That was a typo, a reference got broken, fixed now. Thanks!

---

## Round 2 · Author Response

First of all, thank you for your feedback on this paper.
Indeed, there were a few overlooked issues with the paper (a couple of references missing, device description details lacking), which are revised and fixed now. We improved the literature coverage of IV-VI semiconductor nanowires in the introductory part and added more experimental details about our nanowires and devices, and how they are different from each other.
We comment in subsequent posts in more detail.

You are currently on this page
Sergey Frolov on 2022-05-24 [id 2521]
Referee 2
Response: we expanded our literature overview.
Response: We have now specified these parameters.
Response: We explained this now. There are two types of devices: with a top gate and without, and most measurements are on top-gate devices. They are nominally the same, though the nanowires are different, including their diameters, as well as contact spacings.
Response: Thanks for pointing it out, we reference these papers in the revised manuscript.
Response: We edited the “brief methods” section so there will be more details about our nanowires growth ([100] axis on GaAs[111] substrates and their dimensions.
We also added a comment on differences between our devices and reference supplementary figure S1 that illustrates them.
Fixed, now it specifies that these figures are in the supplementary section.
That was a typo, a reference got broken, fixed now. Thanks!
Sergey Frolov on 2022-05-24 [id 2520]
Referee 1:
Strengths - The authors carry out mesoscopic measurements on PbTe nanowire quantum dots and discuss their potential for quantum devices. PbTe can have strong spin-orbit splitting and has a large g-factor, making it indeed attractive for the application that the authors consider. It is actually surprising that this material has not yet been considered for this research direction. - The strength and anisotropy of the g-factor are measured for the first time in such structures and argue that PbTe can compete with III-Vs. - The type of measurements combined with the analysis that is carried out are novel and can lead to progress and more questions in an important research area. This is especially true given the fact that the dielectric constant of PbTe is high, which has both advantages and disadvantages when it comes to quantum device performance as pointed out by the authors. - A sufficient number of devices have been studied and are reported on.
In quantum dots with Coulomb energy, valley degeneracy is relatively easy to establish, for instance in carbon nanotube quantum dots three small diamonds followed by a large diamond signify two-fold valley degeneracy. In quantum dots with quenched Coulomb interaction, we cannot see this. But we may be able to identify other means of understanding what happens to the valley degree of freedom in these devices in subsequent measurements.
We added this discussion to the manuscript.
Report The authors report mesoscopic transport measurements done on PbTe nanowire quantum dots. The measurements allow the extraction of the spin-orbit coupling strength and the g-factor of PbTe. The measurements are well presented and the results are important as they demonstrate the potential of PbTe compared to III-V materials for semiconductor-superconductor hybrids. This is especially important for quantum computing devices and the search for Majorana modes. I think the manuscript is suitable for this journal and should be considered for publication based on criterion 3 for acceptance: Open a new pathway in an existing or a new research direction, with clear potential for multipronged follow-up work. But before it can be accepted, the authors should address the issues listed below, in the requested changes.
We added a discussion of this point and the references the referee pointed to.
It is more likely that several quantum dots are created in the nanowire. In future measurements, we shall use local gates that would allow us to define quantum dots, including multiple dots, with greater control and distinguish between these possibilities.
We did not perform any measurements at higher temperatures, except for quick 1D bias and gate sweeps for backgate-only devices (5-8) to check IV curves and backgate voltage dependence. There was no study of these nanowires at higher temperatures except for wire resistance at room temperature to choose devices for cooldown.
It is not surprising to see conductance quantization at higher temperatures. In fact it should get more clear once coherence length is suppressed by temperature and conductance fluctuations due to quantum interference are reduced in amplitude. The energy scale of quantized subbands is ~1meV which is 10K.

---

## Editorial Decision

published